# Racial and ethnic disparities in longitudinal trajectories of cardiovascular risk factors in U.S. middle-aged and older adults

**Matthew E. Dupre**[1,2,3,4]*, **Radha Dhingra**[1], **Hanzhang Xu**[4,5,6,7], **Scott M. Lynch**[2,3,4,5], **Qing Yang**[6], **Cassie Ford**[1], **Michael D. Green**[1], **Ying Xian**[8,9], **Ann Marie Navar**[10], **Eric D. Peterson**[10]

1 Department of Population Health Sciences, Duke University, Durham, NC, United States of America, 2 Department of Sociology, Duke University, Durham, NC, United States of America, 3 Duke University Population Research Institute, Durham, NC, United States of America, 4 Center for the Study of Aging and Human Development, Duke University, Durham, NC, United States of America, 5 Department of Family Medicine and Community Health, Duke University, Durham, NC, United States of America, 6 Duke University School of Nursing, Duke University, Durham, NC, United States of America, 7 Health Services and Systems Research, Duke-NUS Medical School, Singapore, Singapore, 8 Department of Neurology, University of Texas Southwestern Medical Center, Dallas, TX, United States of America, 9 Peter O'Donnell Jr. Brain Institute, University of Texas Southwestern Medical Center, Dallas, TX, United States of America, 10 Division of Cardiology, Department of Medicine, University of Texas Southwestern Medical Center, Dallas, TX, United States of America

* matthew.dupre@duke.edu

**Data Availability Statement:** The current study used sensitive health data produced and distributed by the University of Michigan with funding from the

## Abstract

### Background

Racial and ethnic disparities in cardiovascular disease (CVD) risk factors are well-documented. However, racial and ethnic differences in the longitudinal changes among multiple CVD risk factors are unknown.

### Methods

We used prospective cohort data of U.S. adults aged $\geq$50 in the 2006–2016 Health and Retirement Study. Group-based multi-trajectory models characterized age-related trajectories of systolic blood pressure ([BP] mmHg), non-HDL cholesterol (mg/dL), diabetes mellitus (DM), and smoking. Racial and ethnic differences in the multi-trajectory profiles were examined using multinomial logistic regression. Karlson-Holm-Breen methods were used to assess factors contributing to these associations.

### Results

Among 10,292 participants (median age: 61), approximately 32% had an overall favorable profile of CVD risk factors. Compared with non-Hispanic White adults, non-Hispanic Black adults were more likely to exhibit elevated systolic BP with high risks of DM (relative risk ratio [RRR] = 3.36; 95% CI, 2.69–4.21; $P$ < .001) and with low risks of DM (RRR = 3.23; 95% CI, 2.38–4.38; $P$ < .001). Non-Hispanic Black adults were also more likely to exhibit high rates of smoking with and without other co-occurring risk factors. Hispanic adults were most

National Institute on Aging (NIA U01AG009740; https://hrsdata.isr.umich.edu/). The HRS does not allow third-party distribution of its data, and the data used for analysis cannot be shared publicly due to potentially identifying and sensitive participant information. The data used for this study can be obtained by individually-qualified researchers who are registered users (https://hrsdata.isr.umich.edu/user/register) and have obtained an approved Sensitive Data Access Use Agreement with the HRS (https://hrsdata.isr.umich.edu/data-products/sensitive-health/order-form). All data requests and related questions can be directed to: hrsquestions@umich.edu.

**Funding:** This study was funded in part by the National Institute on Aging (NIA; R01AG069938; MED) and an NIA Diversity Supplement (R01AG069938-02S1; MED, MDG). URL: https://www.nia.nih.gov/ The NIA had no role in the design and conduct of the study; collection, management, analysis, and interpretation of the data, preparation, review, or approval of the manuscript, and decision to submit the manuscript for publication.

**Competing interests:** The authors have declared that no competing interests exist.

likely to exhibit high risks of DM with elevated systolic BP (RRR = 1.74; 95% CI, 1.28–2.38; $P < .001$) and without elevated systolic BP (RRR = 1.90; 95% CI, 1.50–2.40; $P < .001$). Education, income, and country-of-origin were significantly associated with the excess CVD risks observed among racial and ethnic minority groups.

## Conclusions

Significant racial and ethnic disparities were observed in trajectories of CVD risk factors in U.S. adults. Social determinants largely contributed to these associations in non-Hispanic Black adults and Hispanic adults.

## Introduction

Cardiovascular disease (CVD) is a leading cause of disability and death that disproportionately affects racial and ethnic minority groups in the United States [1, 2]. The excess burden of CVD among Hispanic and non-Hispanic Black adults compared with non-Hispanic White adults has been largely attributed to underlying disparities in the prevalence and management of the major risk factors [2]. Socioeconomic disadvantages, such as inadequate access to health care, unsafe physical environments, limited education, and unequal employment opportunities predispose these groups to poor health outcomes, including CVD [2]. Clinical guidelines currently recommend the assessment of several major risk factors, including—systolic blood pressure (BP), cholesterol (total and high-density lipoprotein [HDL]), diabetes mellitus (DM), and smoking—to identify those at increased risk of developing CVD [3]. Effective assessment of CVD risks factors is critical to initiating behavioral modification and/or pharmacological therapies to reduce excess CVD in racial and ethnic minority populations [2, 4].

To date, studies have typically focused on racial and ethnic differences in individual risk factors for CVD and have not fully considered the interplay among multiple risk factors [5–7]. Risk factors such as elevated BP, smoking, and diabetes often cluster and interact to influence an individual's CVD risk, particularly in middle and older adulthood [5, 7, 8]. Moreover, the development and progression of these risk factors are not uniform with increasing age and generally evolve over an adult's life course [6, 8, 9]. Therefore, identifying how CVD risk factors co-occur and vary with age and across individuals will provide critical knowledge for targeting preventive care and tailoring risk-reduction strategies in middle-aged and older adults.

The current study used longitudinal data from a U.S. population-based cohort to examine racial and ethnic disparities in age-related trajectories of CVD risk factors in middle- and older-adulthood. Our study had three objectives. First, to characterize the major trajectories of systolic BP, cholesterol, smoking, and diabetes that occurred with increasing age. Second, to examine whether racial and ethnic minority groups were more likely to exhibit distinct age-related trajectories of these CVD risk factors. Third, to identify factors contributing to racial and ethnic disparities in the age-related profiles of CVD risk factors among middle-aged and older adults.

## Materials and methods

### Study participants

The study used nationally representative data from the Health and Retirement Study (HRS). The HRS is sponsored by the National Institute on Aging (grant number NIA U01AG009740)

and is conducted by the University of Michigan [10]. The HRS is the largest ongoing prospective study of U.S. middle-aged and older adults that has interviewed more than 40,000 participants since its launch in 1992 [11, 12]. With biennial interviews and newly added cohorts since 1998, the HRS uses an accelerated longitudinal design that consolidates repeated observations from multiple cohorts to provide robust age-specific data from a nationally-representative sample of adults over age 50 [11, 12]. Beginning in 2006, 50% random half-samples of HRS participants were selected to complete enhanced face-to-face interviews, detailed physical exams, and blood-spot samples [10] every 48 months. Further details of the multistage sampling design, data collection procedures, and response rates have been documented elsewhere [11, 12]. The STROBE guidelines [13] were implemented in planning and conducting this study (S1 Appendix in S1 File).

The current study used data from 31,282 participants who completed detailed interviews and exams and were followed from 2006 to 2016 (data accessed: 06/16/2023). The study was limited to adults aged 50 and older at baseline (n = 30,505) and had complete data on age, sex, race, ethnicity (n = 30,372), and measures of CVD risk factors at baseline (n = 20,398). The analytic sample was further limited to participants who provided 2 or more observations over the study period (n = 10,292). The analyses presented here focus on changes in CVD risk factors up to age 80 to maximize the robustness of the age-specific data and to align with the upper-limit of age used in guideline tools for CVD risk-assessment [14]. The sampling distributions (age, sex, race, ethnicity, and geographic region) of participants in the final analytic sample were consistent with participants in the overall HRS sample. All HRS participants provided written informed consent. The current study was approved by the Institutional Review Board at Duke University Health System (Pro00108869).

## Measures

**Primary outcomes.** This study examined the major risk factors for CVD identified by the American College of Cardiology and American Heart Association (ACC/AHA), including systolic BP, cholesterol (total and HDL-C), smoking, and DM [14]. Systolic BP (mmHg) was ascertained by the average of three blood pressure measurements obtained from the respondent's left arm using a validated automated device [11, 15]. Non-HDL cholesterol (mg/dL) was included as a single measure to facilitate a more parsimonious, yet equally robust indicator of CVD risk associated with dyslipidemia [9, 16]. The self-reported use of anti-hypertensive medications (y/n) and statins (y/n) were also included to account for new or ongoing treatments (detailed below). Current smoking status (y/n) was ascertained from information reported by study participants. The diagnosis of diabetes (y/n) was ascertained based on doctor diagnoses reported by participants and/or values of glycosylated hemoglobin (HbA1C $\geq$ 6.5). The four CVD risk factors (systolic BP, non-HDL cholesterol, smoking, and diabetes) and respective medications (anti-hypertensives and statins) were measured at each wave to provide longitudinal information on age-related changes for each factor [11, 15].

**Race and ethnicity.** Participants' race and ethnicity were ascertained from self-reported interviews and categorized as non-Hispanic Black adults (17.3%), Hispanic adults (13.2%), non-Hispanic White adults (66.5%), or other racial and ethnic group (3.1%).

**Covariates.** Sociodemographic characteristics were assessed at baseline and included sex (male or female), nativity (U.S.-born or foreign-born), marital status (never married, married, divorced, or widowed), educational attainment (less than high school [HS], HS degree, some college, or college/more), and household (HH) income (scaled in $10k dollars). We also included U.S. census region (South vs. non-South) to account for known geographic differences associated with greater CVD risks in the southern United States [17]. Health-related

characteristics were assessed at baseline and included regularly engaging in vigorous physical activity (yes or no), body mass index ([BMI] kg/m$^2$ categorized as underweight, normal, overweight, or obese), alcohol consumption (never, moderate [1–2 drinks/day], or heavy [3 + drinks/day]), number of depressive symptoms measured by the 8-item Center for Epidemiologic Studies Depression Scale (range = 0–8) [18], doctor-diagnosed anxiety/psychiatric disorder (yes or no), and a history of diagnosed (yes or no) heart disease (coronary heart disease, angina, congestive heart failure, heart attack, or other heart problems), stroke (or transient ischemic attack [TIA]), chronic obstructive pulmonary disease (COPD), and cancer (excluding skin cancer) [11, 12].

## Statistical analyses

We calculated the distributions of participants' sociodemographic and health-related characteristics by race and ethnicity using chi-square and Kruskal-Wallis tests for categorical and continuous variables, respectively. We then used group-based multi-trajectory models [19] to examine the major patterns of CVD risk factors that were observed with increasing age. Group-based multi-trajectory models are an extension of group-based trajectory models (GBTM) that allow us to characterize and examine the interrelationships among multiple factors (i.e., systolic BP, non-HDL, etc.) over time [19]. As a data-driven approach, this method makes no assumption about the distribution of trajectories in the population and instead uses statistical criteria to approximate the distribution of multiple trajectories in the population [20, 21]. In doing so, it allowed us to identify groups of adults who exhibited probabilistically similar patterns of CVD risk factors that varied with increasing age. The group-based multi-trajectory models were estimated using the *traj* package in Stata 18.0 (StataCorp LP, College Station, TX) [22].

The distributions for each of the CVD risk factors were assessed in preliminary analyses and showed that censored normal (for systolic BP and non-HDL cholesterol) and logit (for smoking and diabetes) specifications were most appropriate, respectively, to estimate the age-related changes among risk factors. We evaluated group-based multi-trajectory models with up to 10 trajectory groups using linear and quadratic terms to characterize the age-related trajectory curves for the CVD risk factors. The optimal number of trajectory groups was identified based on Bayesian information criteria (BIC), average posterior probabilities (AvePP) of group membership (AvePP values greater than .70 indicate high accuracy in group assignment), and 95% confidence intervals (CI) [23]. The trajectory groups were also assessed for clinical utility based on guideline recommendations [3], over-identification (i.e., redundant risk groups), and under-identification (e.g., identifying a single group for smoking, suggesting that all smokers had similar levels of systolic BP, non-HDL cholesterol, and diabetes).

Time-varying measures for use of anti-hypertensive medications and statins were included in the group-based multi-trajectory models to account for changes in the trajectories of systolic BP and non-HDL cholesterol, respectively, due to new/ongoing treatments [24]. The models also included sex to improve the accuracy of group membership assignment [19]. Finally, preliminary analyses assessed mortality as a modeled component of non-random attrition when classifying the trajectories of CVD risk factors (*dropout* function in traj) [21, 25]. However, overall mortality during the study period was relatively low (6.9%) and the results did not differ.

Multinomial logistic regression models were used to examine the associations between race and ethnicity and trajectory group membership in an unadjusted model (Model 1) and a fully adjusted model (Model 2) that included all sociodemographic and health-related covariates. We then used Karlson-Holm-Breen (KHB) methods to quantify the amount that each covariate attenuated the association between race and ethnicity and trajectory group membership [26–28]. The KHB method is a mediation analysis for nested multinomial models that allowed

us to assess the effects of the covariates (i.e., percent mediating) with their inclusion in Model 2. All analyses were performed using Stata 18.0 (StataCorp LP, College Station, TX). *P* values < 0.05 were considered statistically significant.

## Results and discussion

The median (interquartile range) age of study participants at baseline was 61 (12) years, and the majority were female (59.0%) and non-Hispanic White adults (66.5%) (Table 1). Compared to non-Hispanic White participants, non-Hispanic Black participants and Hispanic participants had lower levels of socioeconomic status and were more likely to be obese, and report greater sedentary behavior, more depressive symptoms, and higher use of statins and antihypertensive medications at baseline. Results from the group-based multi-trajectory models identified 8 groups with distinct age-related changes in CVD risk factors (Fig 1A and 1B). Approximately 32% of adults (Group 1) had an overall favorable profile of CVD risk factors. Group 2 (23.6%) was mainly characterized by moderate-to-high systolic BP; Group 3 (14.8%) had high risks of DM; Group 4 (9.6%) had high rates of smoking; Group 5 (7.7%) had high systolic BP, high risks of DM, and high non-HDL cholesterol; Group 6 (4.7%) had high non-HDL cholesterol and low-to-moderate risks of DM; Group 7 (4.1%) had low-to-moderate levels of systolic BP, moderate-to-high risks of DM, and high levels of non-HDL cholesterol and smoking that declined with age; and Group 8 (3.3%) had excessively high systolic BP, moderate levels of non-HDL cholesterol, and high rates of smoking that declined with age. All groups had a high degree of accuracy in the assignment of group membership (AvePP = .75-.95). Baseline values for the individual risk factors (S1 Table in S1 File) and participants' background characteristics (S2 Table in S1 File) are also shown for each of the trajectory groups.

Table 2 presents the results from the multinomial estimates of the associations between race and ethnicity and trajectory group membership. Results from the unadjusted models showed that non-Hispanic Black adults were much more likely to be in trajectory groups exhibiting elevated systolic BP and non-HDL cholesterol with high risks of DM (Group 5: relative risk ratio [RRR] = 4.05; 95% CI, 3.32–4.94; *P* < .001) and with low risks of DM (Group 8: RRR = 4.59; 95% CI, 3.51–6.02; *P* < .001) compared with non-Hispanic White adults. Non-Hispanic Black adults were also more likely to be in trajectory groups exhibiting high rates of smoking and elevated non-HDL cholesterol with high risks of DM (Group 7: RRR = 4.44; 95% CI, 3.47–5.69; *P* < .001) and with low risks of DM (Group 4: RRR = 2.00; 95% CI, 1.65–2.41; *P* < .001) compared with non-Hispanic White adults. Non-Hispanic Black adults were also more likely to be in Group 3 (RRR = 2.67; 95% CI, 2.27–3.15; *P* < .001) with high risks of DM in the absence of elevated risks for other factors.

Hispanic adults exhibited comparable risks (unadjusted RRRs) for many of the same suboptimal profiles of CVD risk factors as non-Hispanic Black adults (Groups 3, 5, 7, and 8). Unlike non-Hispanic Blacks adults, however, Hispanic adults were significantly more likely to be in Group 6 exhibiting consistently high levels of non-HDL cholesterol and low-to-moderate risks for DM (RRR = 1.51; 95% CI, 1.10–2.07; *P* = .011) compared with non-Hispanic White adults.

The inclusion of covariates in Model 2 (Table 2) partially or fully attenuated the associations between race and ethnicity and trajectory group membership. Table 3 (for non-Hispanic Black adults) and Table 4 (for Hispanic adults) present the results from the KHB analyses showing the percentages that the covariates mediated the racial and ethnic differences in the respective CVD risk profiles. Lower education and income accounted for approximately 26–58% of the likelihood that non-Hispanic Black adults were in groups exhibiting co-occurring risks related to elevated systolic BP, DM, and smoking (Groups 4, 7, and 8) compared with non-Hispanic White adults. Obesity accounted for 17–21% of the likelihood that non-Hispanic Black adults

**Table 1. Baseline characteristics of U.S. middle-aged and older adults by race and ethnicity (n = 10,225).**

| | Overall (n = 10,225) | Non-Hispanic White (n = 6,809) | Non-Hispanic Black (n = 1,759) | Hispanic (n = 1,344) | Non-Hispanic Other (n = 313) | P Value |
|---|---|---|---|---|---|---|
| **Sociodemographic Factors** | | Median (IQR) or n (%) | | | | |
| Age (years) | 61.0 (12.0) | 63.0 (13.0) | 59.0 (11.0) | 58.0 (10.0) | 57.0 (10.0) | < .001 |
| Sex | | | | | | < .001 |
| Male | 4193 (41.0) | 2880 (42.3) | 628 (35.7) | 540 (40.2) | 145 (46.3) | |
| Female | 6032 (59.0) | 3929 (57.7) | 1131 (64.3) | 804 (59.8) | 168 (53.7) | |
| Nativity | | | | | | < .001 |
| Foreign born | 1345 (13.2) | 258 (3.8) | 104 (5.9) | 838 (62.4) | 145 (46.3) | |
| Native born | 8880 (86.9) | 6551 (96.2) | 1655 (94.1) | 506 (37.6) | 168 (53.7) | |
| Region | | | | | | < .001 |
| South region | 4137 (40.5) | 2489 (36.6) | 1028 (58.4) | 522 (38.8) | 98 (31.3) | |
| Non-South region | 6088 (59.5) | 4320 (63.5) | 731 (41.6) | 822 (61.2) | 215 (68.7) | |
| Marital Status | | | | | | < .001 |
| Never married | 484 (4.7) | 228 (3.4) | 187 (10.6) | 55 (4.1) | 14 (4.5) | |
| Married | 7240 (70.8) | 5165 (75.9) | 872 (49.6) | 966 (71.9) | 237 (75.7) | |
| Divorced | 1509 (14.8) | 810 (11.9) | 434 (24.7) | 222 (16.5) | 43 (13.8) | |
| Widowed | 992 (9.7) | 606 (8.9) | 266 (15.1) | 101 (7.5) | 19 (6.1) | |
| Educational Attainment | | | | | | < .001 |
| Less than HS | 1695 (16.6) | 608 (8.9) | 411 (23.4) | 629 (46.8) | 47 (15.0) | |
| HS degree | 3529 (34.5) | 2492 (36.6) | 601 (34.2) | 347 (25.8) | 89 (28.4) | |
| Some college | 2592 (25.4) | 1792 (26.3) | 470 (26.7) | 263 (19.6) | 67 (21.4) | |
| College | 2409 (23.6) | 1917 (28.2) | 277 (15.8) | 105 (7.8) | 110 (35.1) | |
| HH income (per $10k) | 4.7 (6.3) | 5.8 (6.8) | 2.8 (4.3) | 2.6 (3.8) | 5.1 (7.7) | < .001 |
| **Health-Related Factors** | | Median (IQR) or N (%) | | | | |
| Physical inactivity | | | | | | < .001 |
| Sedentary | 5365 (52.5) | 3432 (50.4) | 1043 (59.3) | 738 (54.9) | 152 (48.6) | |
| Non-sedentary | 4860 (47.5) | 3377 (49.6) | 716 (40.7) | 606 (45.1) | 161 (51.4) | |
| Body Mass Index (BMI) | | | | | | < .001 |
| Underweight | 199 (1.9) | 100 (1.5) | 24 (1.4) | 69 (5.1) | 6 (1.9) | |
| Normal weight | 2438 (23.8) | 1816 (26.7) | 300 (17.1) | 222 (16.5) | 100 (31.9) | |
| Overweight | 3794 (37.1) | 2593 (38.1) | 580 (32.9) | 510 (37.9) | 111 (35.5) | |
| Obese | 3794 (37.1) | 2300 (33.8) | 855 (48.6) | 543 (40.4) | 96 (30.7) | |
| Alcohol Consumption | | | | | | < .001 |
| Never | 6264 (61.3) | 3872 (56.9) | 1207 (68.6) | 945 (70.3) | 240 (76.7) | |
| Moderate | 2869 (28.1) | 2220 (32.6) | 391 (22.2) | 213 (15.9) | 45 (14.4) | |
| Heavy | 1092 (10.7) | 717 (10.5) | 161 (9.2) | 186 (13.8) | 28 (8.9) | |
| Depressive symptoms | 1.0 (2.0) | 0.0 (2.0) | 1.0 (3.0) | 1.0 (3.0) | 1.0 (2.0) | < .001 |
| CVD Medications | | | | | | |
| Anti-hypertensives | 4854 (47.5) | 3014 (44.3) | 1017 (62.9) | 598 (44.5) | 135 (43.1) | < .001 |
| Statins | 3921 (38.4) | 2635 (38.7) | 668 (37.9) | 493 (36.7) | 125 (40.0) | .499 |
| Diagnosed Conditions | | | | | | |
| Psychiatric disorder | 1632 (15.9) | 1157 (16.9) | 232 (13.2) | 209 (15.6) | 34 (10.9) | < .001 |
| Heart disease | 1671 (16.3) | 1195 (17.6) | 295 (16.8) | 137 (10.2) | 44 (14.0) | < .001 |
| Stroke or TIA | 474 (4.6) | 286 (4.2) | 119 (6.8) | 54 (4.0) | 15 (4.8) | < .001 |
| COPD | 754 (7.4) | 565 (8.3) | 120 (6.8) | 53 (7.4) | 16 (5.1) | < .001 |

*(Continued)*

**Table 1.** (Continued)

| | Overall (n = 10,225) | Non-Hispanic White (n = 6,809) | Non-Hispanic Black (n = 1,759) | Hispanic (n = 1,344) | Non-Hispanic Other (n = 313) | P Value |
|---|---|---|---|---|---|---|
| Sociodemographic Factors | | Median (IQR) or n (%) | | | | |
| Cancer | 1027 (10.0) | 791 (11.6) | 141 (8.0) | 75 (5.6) | 20 (6.4) | < .001 |

Abbreviations: IQR: interquartile range; HS, high school; HH: household.

Note: Participants with missing data on covariates (n = 67) were dropped.

Abbreviations: IQR: interquartile range; TIA, transient ischemic attack; COPD, chronic obstructive pulmonary disease.

were in groups that had higher age-related risks of DM with elevated systolic BP (Group 5) and without elevated systolic BP (Group 3) relative to non-Hispanic White adults. For Hispanic adults, education and income also played a large role (upwards of 87% and 28%, respectively) in mediating their association with suboptimal trajectories of CVD risk factors. Nativity was another important factor (contributing ~18%) in the likelihood that Hispanic adults were in groups that exhibited high risks for DM with elevated systolic BP (Group 5) and without elevated systolic BP (Group 3) compared with non-Hispanic White adults.

This study is the first prospective investigation of racial and ethnic disparities in age-related trajectories of multiple CVD risk factors among U.S. middle-aged and older adults. Using

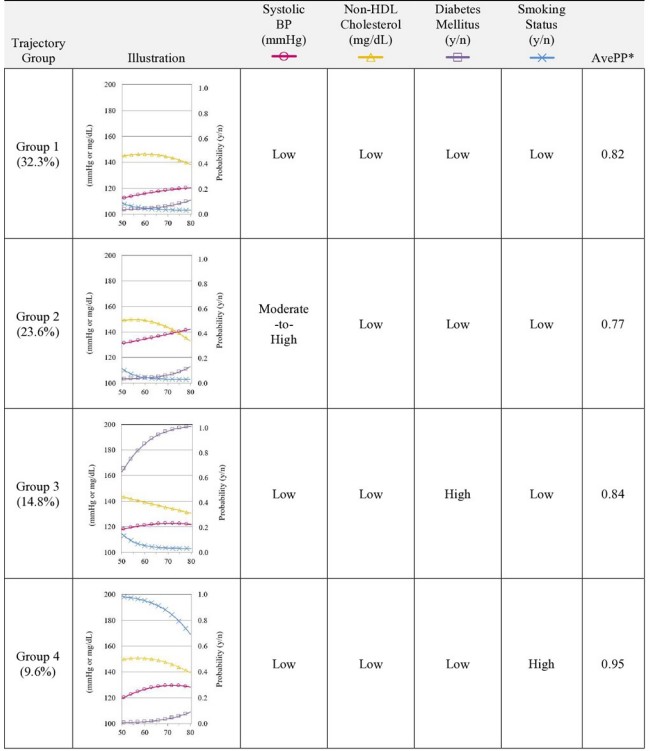
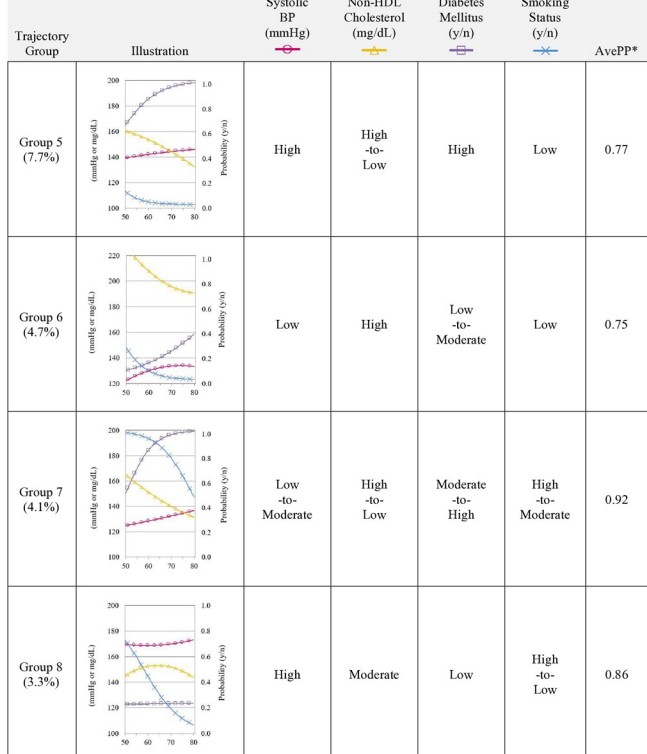

**Fig 1. Estimated trajectory groups characterizing age-related changes in CVD risk factors among U.S. middle-aged and older adults (n = 10,292).**

**Table 2. Multinomial estimates of the association between race and ethnicity and trajectory group membership in U.S. middle-aged and older adults (n = 10,225).**

| | Model 1 (Unadjusted) | | Model 2 (Fully-Adjusted) | |
|---|---|---|---|---|
| | RRR (95% CI) | P | RRR (95% CI) | P |
| Group 2 | | | | |
| Race and Ethnicity | | | | |
| Non-Hispanic White | 1.00 | | 1.00 | |
| Non-Hispanic Black | 1.16 (0.99–1.36) | .068 | 1.23 (1.03–1.47) | .020 |
| Hispanic | 1.09 (0.92–1.30) | .304 | 0.92 (0.73–1.16) | .496 |
| Other race and ethnicity | 0.96 (0.68–1.34) | .800 | 0.97 (0.67–1.41) | .884 |
| Group 3 | | | | |
| Race and Ethnicity | | | | |
| Non-Hispanic White | 1.00 | | 1.00 | |
| Non-Hispanic Black | 2.67 (2.27–3.15) | < .001 | 2.11 (1.76–2.53) | < .001 |
| Hispanic | 2.81 (2.38–3.33) | < .001 | 1.90 (1.50–2.40) | < .001 |
| Other race and ethnicity | 2.41 (1.75–3.33) | < .001 | 2.36 (1.66–3.37) | < .001 |
| Group 4 | | | | |
| Race and Ethnicity | | | | |
| Non-Hispanic White | 1.00 | | 1.00 | |
| Non-Hispanic Black | 2.00 (1.65–2.41) | < .001 | 1.36 (1.09–1.68) | .006 |
| Hispanic | 1.08 (0.85–1.37) | .545 | 0.71 (0.52–0.97) | .032 |
| Other race and ethnicity | 1.05 (0.67–1.67) | .822 | 1.13 (0.69–1.86) | .632 |
| Group 5 | | | | |
| Race and Ethnicity | | | | |
| Non-Hispanic White | 1.00 | | 1.00 | |
| Non-Hispanic Black | 4.05 (3.32–4.94) | < .001 | 3.36 (2.69–4.21) | < .001 |
| Hispanic | 3.02 (2.42–3.77) | < .001 | 1.74 (1.28–2.38) | < .001 |
| Other race and ethnicity | 2.53 (1.66–3.87) | < .001 | 2.52 (1.58–4.02) | < .001 |
| Group 6 | | | | |
| Race and Ethnicity | | | | |
| Non-Hispanic White | 1.00 | | 1.00 | |
| Non-Hispanic Black | 1.32 (0.97–1.82) | .082 | 1.17 (0.83–1.63) | .371 |
| Hispanic | 1.51 (1.10–2.07) | .011 | 1.21 (0.80–1.83) | .358 |
| Other race and ethnicity | 1.46 (0.81–2.66) | .210 | 1.67 (0.89–3.13) | .110 |
| Group 7 | | | | |
| Race and Ethnicity | | | | |
| Non-Hispanic White | 1.00 | | 1.00 | |
| Non-Hispanic Black | 4.44 (3.47–5.69) | < .001 | 2.78 (2.10–3.67) | < .001 |
| Hispanic | 2.39 (1.76–3.25) | < .001 | 1.18 (0.78–1.79) | .429 |
| Other race and ethnicity | 2.89 (1.73–4.85) | < .001 | 2.65 (1.50–4.66) | .001 |
| Group 8 | | | | |
| Race and Ethnicity | | | | |
| Non-Hispanic White | 1.00 | | 1.00 | |
| Non-Hispanic Black | 4.59 (3.51–6.02) | < .001 | 3.23 (2.38–4.38) | < .001 |
| Hispanic | 2.18 (1.54–3.10) | < .001 | 1.23 (0.77–1.94) | .384 |

(*Continued*)

**Table 2.** (Continued)

| | Model 1 | | Model 2 | |
|---|---|---|---|---|
| | (Unadjusted) | | (Fully-Adjusted) | |
| | RRR (95% CI) | *P* | RRR (95% CI) | *P* |
| Other race and ethnicity | 1.53 (0.73–3.22) | .258 | 1.67 (0.77–3.63) | .194 |

Abbreviations: RRR, relative risk ratio; CI, confidence interval.

*Note*: Group 1 (Low CVD risks) is the reference group in the multinomial model. Model 1 is unadjusted. Model 2 included age, sex, nativity, geographic region, marital status, education, household income, inactivity, BMI, alcohol consumption, depressive symptoms, and history of anxiety disorder, heart disease, stroke (or TIA), COPD, or cancer. Participants with missing data on covariates (n = 67) were dropped.

nationally representative data, we identified 8 major groups with distinct age-varying profiles of systolic BP, non-HDL cholesterol, diabetes, and smoking. We also found significant racial and ethnic disparities in the likelihood of exhibiting suboptimal trajectories of one or more CVD risk factors with increasing age. Although these disparities persisted after adjusting for a wide range of factors, results showed that socioeconomic factors played a significant role in the associations between race and ethnicity and the suboptimal trajectories of CVD risk factors compared to baseline comorbidities and other factors.

Our study identified clinically distinct profiles of CVD risk factors that revealed heterogeneity in the presence and/or progression of elevated systolic BP, non-HDL cholesterol, diabetes, and smoking. These findings underscore the importance of considering the complex interplay among multiple risk factors that are observed among U.S. middle-aged and older adults [5–7]. For example, adults with a higher likelihood of being in a trajectory group characterized by elevated systolic BP, elevated non-HDL cholesterol, and higher risks of diabetes (Group 5) would likely benefit from aggressive lifestyle and/or pharmacological interventions to mitigate these compounding risk factors. Although some trajectory groups had a relatively low prevalence in the overall study population (Group 8), these groups identified adults who were at greatest risk and would benefit most from early and aggressive clinical interventions. By recognizing the patterns of high-risk profiles early, clinical decision-making can be tailored to address who may require more intensive prevention and management strategies and multidisciplinary care coordination.

Our study also identified significant racial and ethnic disparities in the age-related trajectories of CVD risk factors [2]. We found that non-Hispanic Black adults were significantly more likely to exhibit suboptimal profiles of co-occurring CVD risk factors compared with non-Hispanic White adults. In particular, non-Hispanic Black adults were much more likely to be characterized with distinct profiles of elevated (and increasing) BP, higher age-related risks of diabetes, and higher probabilities of smoking compared with non-Hispanic White adults. Hispanic adults were especially likely to exhibit higher age-related risks of diabetes with and without elevated BP. Identifying the timing and combinations of these risk factors is vital for informing preventive care efforts and risk-reduction strategies—*e.g.*, when to prioritize lifestyle counseling vs. pharmacological interventions—in racial and ethnic minority populations. Relatedly, integrating socially and culturally-relevant approaches is critical for implementing self-management strategies to reduce CVD risk factors and improving patient outcomes [29–31].

Social determinants of health have gained wide attention for their role in the development, management, and progression of CVD [2]. Our study provides new evidence to support this body of work and demonstrates that socioeconomic factors played a significant role in the associations between race and ethnicity and trajectory group membership [2]. We found that

**Table 3. Percentage of association between race and ethnicity (Non-hispanic black adults) and trajectory group membership attributable to study covariates in U.S. middle-aged and older adults (n = 10,225).**

| | Group 2 vs. Group 1 | Group 3 vs. Group 1 | Group 4 vs. Group 1 | Group 5 vs. Group 1 | Group 6 vs. Group 1 | Group 7 vs. Group 1 | Group 8 vs. Group 1 |
|---|---|---|---|---|---|---|---|
| **Sociodemographic Factors %** | N/A | 6.51 | 67.45 | 3.48 | N/A | 30.91 | 21.06 |
| Age (years) | --- | -1.24 | 10.42 | -2.94 | --- | 6.22 | -2.03 |
| Sex (male) | --- | -5.78 | -9.16 | -6.85 | --- | -5.17 | -5.33 |
| Nativity (foreign born) | --- | 0.70 | -1.75 | 0.54 | --- | -0.26 | -0.23 |
| Region (South) | --- | 2.56 | 1.68 | 0.62 | --- | -1.22 | -0.44 |
| Marital Status (never) | --- | 0.82 | 0.87 | -1.20 | --- | -0.11 | 2.80 |
| Marital Status (divorced) | --- | 1.44 | 7.16 | 0.83 | --- | 1.85 | 1.22 |
| Marital Status (widowed) | --- | 0.17 | 2.60 | 0.47 | --- | 0.61 | 0.76 |
| Education (Less than HS) | --- | 5.25 | 30.65 | 7.07 | --- | 13.98 | 17.61 |
| Education (HS) | --- | -0.41 | -3.09 | -0.83 | --- | -1.59 | -1.80 |
| Education (Some college) | --- | 0.08 | 0.47 | 0.09 | --- | 0.23 | 0.27 |
| HH income (per $10k) | --- | 2.92 | 27.60 | 5.68 | --- | 16.37 | 8.23 |
| **Health-Related Factors %** | N/A | 24.07 | -7.17 | 20.67 | N/A | 10.08 | 6.15 |
| Physical inactivity | --- | 2.01 | 4.47 | 1.70 | --- | 2.35 | 0.77 |
| BMI (underweight) | --- | -0.06 | 0.01 | -0.08 | --- | -0.04 | -0.02 |
| BMI (overweight) | --- | -2.94 | 2.86 | -3.13 | --- | -0.99 | 0.05 |
| BMI (obese) | --- | 20.57 | -16.15 | 17.18 | --- | 4.82 | 2.67 |
| Alcohol consumption (never) | --- | 6.23 | -0.74 | 3.37 | --- | 0.77 | -1.23 |
| Alcohol consumption (heavy) | --- | -0.37 | -2.07 | -0.34 | --- | -0.44 | -0.69 |
| Depressive symptoms | --- | 0.38 | 5.14 | 0.86 | --- | 3.91 | 1.73 |
| Psychiatric disorder | --- | -0.78 | -1.40 | -0.41 | --- | -0.91 | 0.63 |
| Heart disease | --- | -0.21 | 0.31 | -0.09 | --- | -0.14 | 0.18 |
| Stroke or TIA | --- | -0.38 | 0.02 | 0.50 | --- | 0.77 | 1.16 |
| COPD | --- | -0.18 | -1.55 | 0.24 | --- | -0.74 | 0.06 |
| Cancer | --- | -0.20 | 1.93 | 0.87 | --- | 0.72 | 0.84 |
| **Total %** | N/A | 30.58 | 60.28 | 24.15 | N/A | 40.99 | 27.21 |
| *P* value | --- | < .001 | < .001 | < .001 | --- | < .001 | < .001 |

Abbreviations: HS, high school; HH, household; BMI, body mass index; TIA, transient ischemic attack; COPD, chronic obstructive pulmonary disease.

*Note*: Percentages reported for non-Hispanic Black adults (vs. non-Hispanic White adults) while adjusting for Hispanic and other reported racial groups. Columns with N/A indicate associations that were not statistically significant or not attenuated with the inclusion of covariates. Participants with missing data on covariates (n = 67) were dropped.

* *P* values indicate the effect of sociodemographic and health-related factors on the association between race and trajectory group membership.

adjusting for adults' sociodemographic background accounted for a large portion of the excess CVD risks observed in non-Hispanic Black adults (up to 67%) and Hispanic adults (up to 70%). In particular, we found that lower education and income accounted for about 26–58% of the likelihood of non-Hispanic Black adults exhibiting co-occurring risks related to elevated BP, diabetes, and smoking compared with non-Hispanic White adults. For Hispanic adults, lower education and income also played a large role (28–87%) in their association with suboptimal trajectories of CVD risk factors. Place-of-birth was another key factor in the likelihood that Hispanic adults (born outside the U.S.) exhibited higher-risk CVD profiles relative to non-Hispanic White adults [32]. Among health-related factors, obesity and depressive

**Table 4. Percentage of association between race and ethnicity (Hispanic adults) and trajectory group membership attributable to study covariates in U.S. middle-aged and older adults (n = 10,225).**

| | Group 2 vs. Group 1 | Group 3 vs. Group 1 | Group 4 vs. Group 1 | Group 5 vs. Group 1 | Group 6 vs. Group 1 | Group 7 vs. Group 1 | Group 8 vs. Group 1 |
|---|---|---|---|---|---|---|---|
| **Sociodemographic Factors %** | N/A | 28.17 | N/A | 34.63 | 22.34 | 69.74 | 60.21 |
| Age (years) | --- | -1.72 | --- | -5.40 | -6.34 | 14.33 | -5.66 |
| Sex (male) | --- | -1.73 | --- | -2.72 | -2.70 | -2.58 | -3.21 |
| Nativity (foreign born) | --- | 17.99 | --- | 18.23 | -3.57 | -11.12 | -12.12 |
| Region (South) | --- | 0.25 | --- | 0.08 | 0.48 | -0.20 | -0.09 |
| Marital Status (never) | --- | 0.08 | --- | -0.15 | -0.53 | -0.02 | 0.54 |
| Marital Status (divorced) | --- | 0.48 | --- | 0.37 | -0.38 | 1.04 | 0.83 |
| Marital Status (widowed) | --- | -0.04 | --- | -0.13 | -0.05 | -0.21 | -0.32 |
| Education (Less than HS) | --- | 12.82 | --- | 22.89 | 53.31 | 56.85 | 86.58 |
| Education (HS) | --- | -1.70 | --- | -4.54 | -13.37 | -10.94 | -15.00 |
| Education (Some college) | --- | -1.31 | --- | -1.86 | -4.20 | -5.87 | -8.64 |
| HH income (per $10k) | --- | 3.05 | --- | 7.86 | -0.31 | 28.46 | 17.30 |
| **Health-Related Factors %** | N/A | 16.45 | N/A | 22.51 | 35.57 | 15.26 | 15.97 |
| Physical inactivity | --- | 0.95 | --- | 1.06 | 0.55 | 1.84 | 0.73 |
| BMI (underweight) | --- | 1.97 | --- | 3.34 | -3.06 | 2.41 | 1.38 |
| BMI (overweight) | --- | -0.07 | --- | -0.10 | -0.20 | -0.04 | 0.00 |
| BMI (obese) | --- | 8.55 | --- | 9.48 | 14.25 | 3.34 | 2.24 |
| Alcohol consumption (never) | --- | 6.64 | --- | 4.76 | 5.72 | 1.37 | -2.63 |
| Alcohol consumption (heavy) | --- | 0.82 | --- | 1.00 | 2.76 | 1.65 | 3.13 |
| Depressive symptoms | --- | 0.52 | --- | 1.55 | 11.39 | 8.85 | 4.73 |
| Psychiatric disorder | --- | -0.28 | --- | -0.19 | -0.95 | -0.54 | 0.45 |
| Heart disease | --- | -1.86 | --- | -1.04 | 5.46 | -2.02 | 3.12 |
| Stroke or TIA | --- | 0.02 | --- | -0.04 | 0.04 | -0.08 | -0.16 |
| COPD | --- | -0.49 | --- | 0.89 | -0.08 | -3.40 | 0.35 |
| Cancer | --- | -0.32 | --- | 1.80 | -0.31 | 1.88 | 2.63 |
| **Total %** | N/A | 44.62 | N/A | 57.14 | 57.91 | 85.00 | 76.18 |
| *P* value | --- | < .001 | --- | < .001 | .005 | < .001 | < .001 |

Abbreviations: HS, high school; HH, household; BMI, body mass index; TIA, transient ischemic attack; COPD, chronic obstructive pulmonary disease.

*Note.* Percentages reported for Hispanic adults (vs. non-Hispanic White adults) while adjusting for non-Hispanic Black and other reported racial groups. Columns with N/A indicate associations that were not statistically significant or not attenuated with the inclusion of covariates. Participants with missing data on covariates (n = 67) were dropped.

* *P* values indicate the effect of sociodemographic and health-related factors on the association between ethnicity and trajectory group membership.

symptoms contributed significantly in their association with suboptimal trajectories of CVD risk factors in both non-Hispanic Black adults (up to 24%) and Hispanic adults (up to 36%). Taken together, these results suggest that inadequate health literacy, financial resources, and/or access to health insurance (particularly among non-native groups) may hinder the adoption of a healthy lifestyle and/or the resources necessary to reduce the risks for CVD. To help mitigate these preventable risk factors, healthcare providers should remain vigilant in monitoring racial and ethnic minority groups—especially those who identify as Hispanic, foreign-born (or non-native origin), and those with lower educational attainment and income—for initiating timely and tailored interventions based on their risk profile.

We acknowledge several limitations of this study. First, this study used a data-driven method, and the results may not fully characterize the age-related changes in CVD risk factors for all middle-aged and older adults. Therefore, we encourage studies to further assess the patterns of additional (or different) trajectories of CVD risk factors that may be observed in other datasets/populations. Second, the HRS data are limited in the number of data points available to assess changes in BP, cholesterol, diabetes, and smoking. However, the follow-up data that we used from 2006–2016 provided measures every 4 years for all study subjects—which is consistent with guidelines recommending the use of follow-up measures every 4–6 years when available [3, 14]. In addition, although we accounted for the use of anti-hypertensives and statins over time, we could not differentiate whether age-related changes in BP and cholesterol were due to the initiation/use of these therapies or other factors. Third, although overall mortality during the study period was low (6.9%), we acknowledge that selective survival may have introduced bias in the results. Fourth, there may be additional factors (*e.g.*, diet, residential location, neighborhood-level socioeconomic disadvantage) that may have contributed to the racial and ethnic disparities in the trajectories of CVD risk factors. Additionally, information on medical history was based on participant self-report which may have introduced recall bias and could also indicate a greater access to healthcare services/health insurance coverage among these participants. Nevertheless, the covariates included in our analysis accounted for a sizeable portion (~25–85%) of the excess risks observed in non-Hispanic Black and Hispanic adults. Finally, the study's observational design prevented any causal interpretations.

## Conclusions

In summary, this study was the first to characterize the concurrent longitudinal trajectories of multiple CVD risk factors in a sample of U.S. middle-aged and older adults. Our findings demonstrated heterogeneity in the interrelationships among risk factors and provided new insights to better target preventive care and risk-reduction strategies in racial and ethnic minority populations. In addition, the study highlighted the underlying role of social determinants in CVD risks and the importance of developing multi-level interventions to address the socioeconomic inequities that perpetuate racial and ethnic disparities in CVD. Additional studies, particularly among racial and ethnic minority populations are needed to confirm our findings.

## Supporting information

**S1 File.**
(PDF)

## Author Contributions

**Conceptualization:** Matthew E. Dupre.

**Data curation:** Radha Dhingra, Cassie Ford.

**Formal analysis:** Matthew E. Dupre, Radha Dhingra.

**Funding acquisition:** Matthew E. Dupre, Michael D. Green.

**Investigation:** Matthew E. Dupre, Hanzhang Xu, Scott M. Lynch, Qing Yang, Cassie Ford, Michael D. Green, Ying Xian, Ann Marie Navar, Eric D. Peterson.

**Methodology:** Matthew E. Dupre, Radha Dhingra, Scott M. Lynch, Qing Yang.

**Software:** Matthew E. Dupre, Radha Dhingra, Cassie Ford.

**Writing – original draft:** Matthew E. Dupre.

**Writing – review & editing:** Matthew E. Dupre, Radha Dhingra, Hanzhang Xu, Scott M. Lynch, Qing Yang, Cassie Ford, Michael D. Green, Ying Xian, Ann Marie Navar, Eric D. Peterson.

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
