## [Decision Letter · Decision Letter 0]

4 Dec 2024

PONE-D-24-48343Racial and Ethnic Disparities in Longitudinal Trajectories of Cardiovascular Risk Factors in U.S. Middle-Aged and Older AdultsPLOS ONE

Dear Dr. Dupre,

Thank you for submitting your manuscript to PLOS ONE. After careful consideration, we feel that it has merit but does not fully meet PLOS ONE’s publication criteria as it currently stands. Therefore, we invite you to submit a revised version of the manuscript that addresses the points raised during the review process. The revised version should address all comments. Please submit your revised manuscript by Jan 16 2025 11:59PM. If you will need more time than this to complete your revisions, please reply to this message or contact the journal office at plosone@plos.org. Please include the following items when submitting your revised manuscript:A rebuttal letter that responds to each point raised by the academic editor and reviewer(s). You should upload this letter as a separate file labeled 'Response to Reviewers'.A marked-up copy of your manuscript that highlights changes made to the original version. You should upload this as a separate file labeled 'Revised Manuscript with Track Changes'.An unmarked version of your revised paper without tracked changes. You should upload this as a separate file labeled 'Manuscript'.

We look forward to receiving your revised manuscript.

Kind regards,

Petri Böckerman

Academic Editor

PLOS ONE

2. We note that you have indicated that there are restrictions to data sharing for this study. For studies involving human research participant data or other sensitive data, we encourage authors to share de-identified or anonymized data. However, when data cannot be publicly shared for ethical reasons, we allow authors to make their data sets available upon request. For information on unacceptable data access restrictions, please see http://journals.plos.org/plosone/s/data-availability#loc-unacceptable-data-access-restrictions. Before we proceed with your manuscript, please address the following prompts: a) If there are ethical or legal restrictions on sharing a de-identified data set, please explain them in detail (e.g., data contain potentially identifying or sensitive patient information, data are owned by a third-party organization, etc.) and who has imposed them (e.g., a Research Ethics Committee or Institutional Review Board, etc.). Please also provide contact information for a data access committee, ethics committee, or other institutional body to which data requests may be sent. b) If there are no restrictions, please upload the minimal anonymized data set necessary to replicate your study findings to a stable, public repository and provide us with the relevant URLs, DOIs, or accession numbers. Please see http://www.bmj.com/content/340/bmj.c181.long for guidelines on how to de-identify and prepare clinical data for publication. For a list of recommended repositories, please see https://journals.plos.org/plosone/s/recommended-repositories. You also have the option of uploading the data as Supporting Information files, but we would recommend depositing data directly to a data repository if possible. Please update your Data Availability statement in the submission form accordingly.

Additional Editor Comments:

The revised version should address all comments.

Reviewers' comments:

Reviewer's Responses to Questions

**Comments to the Author**

1. Is the manuscript technically sound, and do the data support the conclusions?

Reviewer #1: Yes

Reviewer #2: Yes

Reviewer #3: Yes

2. Has the statistical analysis been performed appropriately and rigorously? 

Reviewer #1: Yes

Reviewer #2: Yes

Reviewer #3: Yes

3. Have the authors made all data underlying the findings in their manuscript fully available?

Reviewer #1: Yes

Reviewer #2: No

Reviewer #3: Yes

4. Is the manuscript presented in an intelligible fashion and written in standard English?

Reviewer #1: Yes

Reviewer #2: Yes

Reviewer #3: Yes

5. Review Comments to the Author

Reviewer #1: This is a carefully thought-out analyses and the authors were successfully able to draw new data that highlights covariates that mediate disparities of racial and ethnic minorities (in middle-aged and older adults) in relation to their risks for different CVD risk factors, placing them in different trajectory group membership. Most of key data was highlighted and the methods were clear. The study is of significance to the broader scientific community.

I only have small suggestions here. Tables 4 and 5 were mostly utilized to demonstrate the sociodemographic factors that would place Hispanic adults or non-Hispanic black adults in different group memberships. It seems health related factors were not really mentioned and discussed much in text, whether in results / discussion. It might be more suited for discussion. What is the relationship between the health-related factors and the sociodemographic factors (if any)? Or simply acknowledge that looking at Table 4 and Table 5, health-related factors can account for some CVD risks, 36% for Hispanic adults, and 25% for non-Hispanic black adult.

Along the lines of the previous paragraph, can the study comment on baselines health characteristics of non-Hispanic black or Hispanic minorities compared to non-Hispanic white adults? These does not have to necessarily be mentioned with regards to CVD risks, but it would be important as health-related factors are also important covariates to consider in such analyses, whether in context of this study or previous studies.

Reviewer #2: Please expand the introduction section to shed light on the ethnicity and racial background as a determinant for population health, why certain ethnic/ racial groups are considered a health disparities.

Define the rationale of the study

Identify the impact of the study at the population and clinical practice levels.

Because the SD in the age is pretty wide which indicates presence of outliers, please use the median instead.

Define "South" and explain why South had been a classification category.

Expand the discussion section to further correlated between findings of this study and previous studies that had assessed similar population.

Add a section for future recommendations.

Reviewer #3: Thank you for the opportunity to review this manuscript in which the authors elucidate racial and ethnic disparities in longitudinal trajectories of cardiovascular risk factors in middle-aged and older adults in the United States.

This is an exceptional manuscript. The study is well conceived, well executed, and well reported. I just have some minor comments for consideration for this or future work.

1. How might geographic location of participant residential address play a part, for example poor vs wealthy suburbs, and rurality? These play significant roles in access to health care and therefore health outcomes. I believe that the United States has an area deprivation index – I wonder if this might help address this aspect, or if there is a geographic classification for health?

2. Similarly, consideration of participant visa/residency status may shed further light on access to health care and therefore health outcome.

3. For several variables, data is based on participant self-report of physician diagnosis. This presumes access to health provider, health literacy, acceptability of diagnosis. Some discussion of the direction and magnitude of any influence on the results this might have is warranted in the Strengths & Limitations section.

4. Very minor, in the Abstract: missing ‘[to] exhibit high rates of smoking’, and Nativity would benefit from explanation – it is explained in the Introduction but it was a novel term to come across in the abstract.

6. PLOS authors have the option to publish the peer review history of their article (what does this mean?). If published, this will include your full peer review and any attached files.

Reviewer #1: No

Reviewer #2: **Yes: **WEAM BANJAR

Reviewer #3: **Yes: **Dr Miranda Buhler

---

## [Author Response · Author response to Decision Letter 0]

16 Dec 2024

Our responses to reviewer comments are attached

---

## [Decision Letter · Decision Letter 1]

16 Jan 2025

Racial and ethnic disparities in longitudinal trajectories of cardiovascular risk factors in U.S. middle-aged and older adults

PONE-D-24-48343R1

Dear Dr. Dupre,

We’re pleased to inform you that your manuscript has been judged scientifically suitable for publication and will be formally accepted for publication once it meets all outstanding technical requirements.

Kind regards,

Petri Böckerman

Academic Editor

PLOS ONE

Additional Editor Comments (optional):

I am happy with the revised version of the paper.

Reviewers' comments:

Reviewer's Responses to Questions

**Comments to the Author**

1. If the authors have adequately addressed your comments raised in a previous round of review and you feel that this manuscript is now acceptable for publication, you may indicate that here to bypass the “Comments to the Author” section, enter your conflict of interest statement in the “Confidential to Editor” section, and submit your "Accept" recommendation.

Reviewer #1: All comments have been addressed

Reviewer #2: All comments have been addressed

Reviewer #3: All comments have been addressed

2. Is the manuscript technically sound, and do the data support the conclusions?

Reviewer #1: Yes

Reviewer #2: Yes

Reviewer #3: Yes

3. Has the statistical analysis been performed appropriately and rigorously? 

Reviewer #1: Yes

Reviewer #2: (No Response)

Reviewer #3: Yes

4. Have the authors made all data underlying the findings in their manuscript fully available?

Reviewer #1: Yes

Reviewer #2: No

Reviewer #3: Yes

5. Is the manuscript presented in an intelligible fashion and written in standard English?

Reviewer #1: Yes

Reviewer #2: Yes

Reviewer #3: Yes

6. Review Comments to the Author

Reviewer #1: All comments have been addressed and I suggest for the manuscript to be accepted for publication, the study is a unique study and is likely to be of broad interest.

Reviewer #2: Thank you for addressing reviewers concerns, please define restrictions applied on data sharing and highlight protection applied to vulnerable population

Reviewer #3: Thank you to the authors for their responses to the reviewer comments and for their appropriate amendments. I have no further comments to add.

7. PLOS authors have the option to publish the peer review history of their article (what does this mean?). If published, this will include your full peer review and any attached files.

Reviewer #1: No

Reviewer #2: **Yes: **WEAM BANJAR

Reviewer #3: **Yes: **Miranda Buhler

---

## [Editor Report · Acceptance letter]

24 Jan 2025

PONE-D-24-48343R1 

PLOS ONE

Dear Dr. Dupre, 

I'm pleased to inform you that your manuscript has been deemed suitable for publication in PLOS ONE. Congratulations! Your manuscript is now being handed over to our production team.

Kind regards, 

on behalf of

Professor Petri Böckerman 

Academic Editor

PLOS ONE
